# Delirium after Cardiac Surgery—A Narrative Review

**DOI:** 10.3390/brainsci13121682

**Published:** 2023-12-07

**Authors:** Daniel Mattimore, Adrian Fischl, Alexa Christophides, Jerry Cuenca, Steven Davidson, Zhaosheng Jin, Sergio Bergese

**Affiliations:** Department of Anesthesiology, Stony Brook University Hospital, Stony Brook, NY 11794, USA; daniel.mattimore@stonybrookmedicine.edu (D.M.); adrian.fischl@stonybrookmedicine.edu (A.F.); alexa.christophides@stonybrookmedicine.edu (A.C.); jerry.cuenca@stonybrookmedicine.edu (J.C.); steven.davidson@stonybrookmedicine.edu (S.D.); zhaosheng.jin@stonybrookmedicine.edu (Z.J.)

**Keywords:** postoperative delirium, cardiac surgery, cardiopulmonary bypass

## Abstract

Postoperative delirium (POD) after cardiac surgery is a well-known phenomenon which carries a higher risk of morbidity and mortality. Multiple patient-specific risk factors and pathophysiologic mechanisms have been identified and therapies have been proposed to mitigate risk of delirium development postoperatively. Notably, cardiac surgery frequently involves the use of an intraoperative cardiopulmonary bypass (CPB), which may contribute to the mechanisms responsible for POD. Despite our greater understanding of these causative factors, a substantial reduction in the incidence of POD remains high among cardiac surgical patients. Multiple therapeutic interventions have been implemented intraoperatively and postoperatively, many with conflicting results. This review article will highlight the incidence and impact of POD in cardiac surgical patients. It will describe some of the primary risk factors associated with POD, as well as anesthetic management and therapies postoperatively that may help to reduce delirium.

## 1. Introduction

Despite the high burden of comorbidities of many cardiac surgical patients, mortality rates have continued to decline [1]. Nevertheless, postoperative neurocognitive complications and postoperative delirium remain a major concern for these patients. In the era of the cardiopulmonary bypass (CPB), systemic and neuroinflammation, pathologic stress responses, and changes in synaptic function are a few of the pathophysiologic mechanisms thought to underlie these complications. Temperature management, the impact of non-pulsatile flow, and ischemia-reperfusion are also important contributors [2].

Post-operative delirium is defined as fluctuations in attention and awareness that appear within days after surgery. Post-operative cognitive dysfunction, on the other hand, is defined as new cognitive deficits arising immediately after surgery and lasting up to 6 months, with memory impairment and impaired performance on intellectual tasks being the most common manifestations [3]. Post-operative delirium and post-operative cognitive dysfunction are historically thought of as distinct disorders; however, similarities in their mechanisms, risk factors, and long-term consequences allow them to be grouped into what are referred to as perioperative neurocognitive disorders [2].

The etiology of neurocognitive disorders after CPB is multifactorial, and results from a combination of preoperative, intraoperative, and postoperative insults, along with established patient risk factors [4]. Preoperative risk stratification is essential to the management of postoperative neurocognitive complications. Older age, for example, is an independent risk factor for delirium in this population, and the presence of other comorbidities common in most cardiac patients may also increase the risk. Obtaining an appropriate history and physical examination can identify the patients at the highest risk, and this information can then be communicated to the intraoperative and postoperative teams to assist with the targeted surveillance and management of patients. Intraoperatively, certain factors have been proposed to increase or predict the risk of post-operative neurocognitive disorders. These include the CPB time, surgical complexity, and use of a cerebral oximetry or bispectral index (BIS). Postoperatively, non-pharmacological interventions such as reorientation, good sleep hygiene, early mobility, visual and hearing optimization, adequate nutrition, and cognitive exercise have been shown to reduce rates of postoperative delirium. Pharmacologic agents such as antipsychotics have been proposed as a rescue treatment to mitigate postoperative delirium in this population; however, data to support its widespread use are limited [4].

The first 30 days after cardiopulmonary bypass have the most significant effects on a cardiac patient’s quality of life; therefore, early recovery after surgery (ERAS), defined as a widespread protocol of perioperative interventions designed to improve outcomes, are essential in preventing postoperative neurocognitive complications [5].

### Search Strategy

Authors searched and selected articles by use of PubMed, EMBASE, and Google Scholar. Search item descriptions included the following: postoperative cognitive delirium, cardiac surgery, cardiopulmonary bypass, and postoperative cognitive dysfunction. Articles considered for review included clinical literal reviews, basic research reports, clinical trials, case reports, and systemic reviews, and meta-analyses were included if appropriate. References were selected based on relevance to the topic, hierarchy of clinical evidence, and subsequently by study design. The findings from the literature are summarized as a narrative review focused on describing the pathophysiology of POD after cardiac surgery and development of interventions for the prevention and treatment of POD.

## 2. Intraoperative Cardiopulmonary Bypass

Intraoperative cardiopulmonary bypass (CPB), a form of extracorporeal circulation, provides circulatory and respiratory support during cardiac surgery. This physiological support, along with temperature regulation, has made increasingly complex surgery on the heart and great vessels possible in the modern era [6]. During cardiopulmonary bypass, blood drained from the right atrium is diverted through a venous line of the CPB circuit into a reservoir. An arterial pump draws blood from this reservoir and propels it through a heat exchanger, an oxygenator, and finally an arterial line filter. The blood is then returned to the body via a cannula positioned in the ascending aorta or other major artery [7]. Despite the efforts of strict protocols and a standard sequence of events coordinated by anesthesiologists, perfusionists, and surgeons, the prevention of neurological damage and cognitive impairment following CPB remains a challenge.

The primary way CPB contributes to POD may be from the generation of cerebral emboli. Emboli can manifest as either atherosclerotic calcium deposits dislodged during aortic arch cross-clamping or as gaseous microemboli arising from entrapped air within the CPB circuit [8,9]. Strategies exist to mitigate these potential insults. Air micro-emboli may be caused by the manipulation of the CPB circuit, such as the administration of medications or blood draws [10]. Various designs of oxygenators and arterial line filters in the CPB circuit have been developed, in part to reduce air emboli [11]. Epiaortic ultrasound scanning of the aortic arch and root is used to assess for the presence and severity of atherosclerotic plaque prior to surgical aortic manipulation. The use of epiaortic ultrasound scanning allows surgeons to determine whether and where to perform aortic cross clamping for CPB [12]. 

Anticoagulation is necessary to avoid thrombus formation during CPB, for which unfractionated heparin is most commonly used. Inadequate anticoagulation during CPB, defined as an activated clotting time (ACT) of less than 400 s, may significantly increase the risk of acute thrombus formation. However, heparin is known to have possible unwanted effects such as heparin resistance, heparin-induced thrombocytopenia, or allergic reactions [13]. In these cases, other anticoagulants have been utilized, such as bivilrubin and nafamostat mesylate. 

Nafamostat mesylate is a short-acting protease inhibitor which directly inhibits thrombin, thus prolong ACT. In a retrospective observational study involving over 800 patients who received nafamostat mesylate, not only were appropriate ACT measurements attained for CPB, but there was also no evidence of increased perioperative ischemic stroke [14].

Bivalirudin is a direct thrombin inhibitor most commonly used in patients undergoing percutaneous coronary interventions. Its use has also expanded to anticoagulation during CPB as well as extracorporeal membrane oxygenation. While case reports have demonstrated effectiveness in achieving targeted ACT levels during CPB without an increase in adverse effects such as bleeding or thrombotic events [15], more research is needed to assess bivalirudin and other newer-generation anticoagulation efficacies in CPB [16].

## 3. Postoperative Delirium in Cardiac Surgery 

No exact pathophysiological change responsible for POD is known, but there are multifaceted etiologic factors contributing to its occurrence. In cardiac surgery, factors such as systemic inflammation, hypothermia, non-pulsatile flow from CPB, and ischemic-reperfusion injury may all contribute to POD [17,18].

Surgery elicits an inflammatory response targeting not just the surgical field, but also neurologic tissue. The release of interleukin 6 (IL-6) and C-reactive protein is involved in the inflammatory cascade and leads to the increased permeability of the blood–brain-barrier, which is responsible for development of cerebral edema [17,19].

There is conflicting evidence surrounding whether intraoperative hypothermia common with CPB is associated with POD. In a prospective randomized study for patients undergoing aortic valve replacement, normothermic temperature management during CPB was noninferior to hypothermia in the development of postoperative neurocognitive decline [20]. However, there is evidence that mild hypothermia during CPB (defined as 32 °C) may lead to worst neurocognitive outcomes compared to normothermic CPB (defined as 37 °C) during CBP [21].

During CPB, blood is either circulated in a pulsatile or non-pulsatile manner. Pulsatile blood flow mimics normal physiologic blood flow, but there is controversy as to whether any significant benefit exists. Evidence suggests it enhances microcirculatory blood flow and tissue perfusion, but no demonstration of superiority to non-pulsatile blood flow on outcomes has been made [18,22]. Evidence also suggests that non-pulsatile blood flow increases carotid artery wall stress, and can lead to shedding of the endothelial glycocalyx—a process also involved in systemic inflammation and perfusion–reperfusion injury [23].

The molecular mechanisms of ischemia–reperfusion injury have been well studied in hopes that targeting these pathways with new therapies may reduce cerebral injury. Without a ready supply of oxygen and glucose, ATP is quickly depleted, prolonged cell membrane depolarization occurs, and a cascade of molecular events ultimately leads to cell apoptosis [24,25]. The reperfusion of affected cells may hasten cell death by introducing oxygen-derived free radicals, disrupting the blood–brain barrier and increasing inflammatory mediators such as cytokines, TNF-alpha, IL-1 beta, and IL-6 [25,26].

The incidence of postoperative cognitive dysfunction varies between studies, from as low as 11% at 3 months after cardiac surgery [27] to 41.4% at the time of discharge from the hospital [28]. Postoperative delirium can also be a contributing factor to postoperative cognitive dysfunction. The difference between the two is that delirium manifests shortly after surgery and is described as rapid onset episodes of disorganized, thoughts, behavior, and confusion without causation by drugs, infection, or metabolic disorder [29]. The incidence is reported to be as high as over 50% of patients who undergo cardiac surgery [30]. Cognitive dysfunction, on the other hand, is generally understood to exist 30 days to 1 year or more after surgery, and can be diagnosed with a multitude of neuropsychologic diagnostic tests such as the Mini–Mental State Examination (MMSE), Cognitive Failure Questionnaire (CFQ), or Confusion Assessment Method (CAM) [31]. Postoperative cognitive delirium is associated with worse outcomes, such as roughly tenfold higher mortality, increased length of hospital stay, increased cost, infectious complications, and postoperative cognitive dysfunction [32].

## 4. Management of Postoperative Delirium in the Cardiac Surgery Population

Brain injury is a major factor of postoperative morbidity after CPB, and is associated with prolonged hospitalization, excessive operative mortality, increased hospital costs, and decreased quality of life [30]. Multimodal magnetic resonance imaging following CPB demonstrates potential brain infarction in up to 45% of patients following cardiac surgical interventions [33,34]. It is postulated that cerebral embolism, hypoperfusion secondary to inflammatory process from CBP, or reperfusion injury are all possible contributors to brain injury after cardiac surgery [34]. Although these mechanisms have not been extensively evaluated, they pose possible long-term adverse outcomes, especially as these patients are inherently prone to the development of cerebrovascular disease. It is, therefore, prudent to adhere to perioperative cerebrovascular preventative measures in order to mitigate cerebral hypoperfusion [35].

Many anesthesiologists employ evidence-based recommendations targeted at reducing perioperative cerebral insult. For example, blood pressure management during CPB is typically kept at a minimum mean arterial pressure of 50 mmHg to maintain adequate cerebral blood flow, ensuring adequate cerebral oxygenation and nutrient delivery [36]. Membrane oxygenators are gas-permeable membranes used for CO_2_ extraction and O_2_ delivery to blood. Arterial line filters aid in capturing embolic material which would otherwise potentially result in cerebral microinfarctions [34]. Epiaortic ultrasounds used for the detection of atherosclerosis of the ascending aorta help guide surgeons on manipulation of aorta when considering reducing stroke risk [34].

Comprehensive preoperative evaluation regarding interdisciplinary communication and a multidisciplinary approach is necessary for the optimization of the cardiac patient population. Many of these patients have multiple comorbidities, such as coronary artery disease, valvular heart disease, aneurysms, cardiac arrhythmias, and kidney disease, which pose a risk to their perioperative course. This patient population, additionally, is traditionally made up of the older population. A study in 2007 in the United Kingdom found that of the 25,000 coronary artery bypass graft (CABG) surgeries performed each year, almost 25% of the patients are >70, and 8% are >75 [37]. Most institutions focus on the preoperative optimization of these patients in terms of alcohol and smoking cessation, blood glucose control, nutritional deficiency correction, avoidance of prolonged fasting, increased use of patient engagement technology, and comprehensive rehabilitation [5]. The identification of cognitive issues, substance and alcohol use disorder, history of mood disorders, and frailty of patients should additionally be considered for targeted surveillance and management during the perioperative course [4].

### 4.1. Preoperative Interventions

Delirium is closely associated with increased mortality and duration of hospitalization and prolonged cognitive dysfunction. Unfortunately, the exact pathophysiology continues to elude us. A thorough pre-operative patient assessment provides a high-quality risk stratification for the development of delirium after cardiac surgery. However, patients undergoing cardiac surgery often have several non-modifiable risk factors, including advanced age, hearing/visual impairments, history of cerebrovascular or pulmonary disease, and dementia [38]. The pre-operative optimization of patients can help reduce the incidence of POD.

A recent metanalysis of over 60,000 patients who underwent CABG noted that the presence of modifiable risks factors is correlated with the development of POD. They found that the presence of preoperative depression, diabetes mellitus and hypertension, peripheral vascular disease, and BMI > 30 kg/m^2^ in patients increased the risk of POD [39]. In non-cardiac surgery, a meta-analysis including over 8000 patients concluded that those of advanced age (65 years and older) were at the highest risk of developing POD [40]. Other independent risk factors identified for the development of POD were smoking and the male gender. Interestingly, a higher level of education attainment was associated with over a 50% reduction in the development of POD. Addressing modifiable risk factors may aid in reducing the risk of POD [40].

There have been initiatives in the geriatric patient population (age 65 years and older) to tailor preoperative medication use in order to mitigate POD. Pre-operative comprehensive geriatric assessment (CGA) is used to assess patients’ medications, functional status, nutritional status, cognitive function, and possible underlying socioeconomic issues. In one study of 475 geriatric patients undergoing cancer surgery, several medications were identified in routine CGA that were highly correlated with the development of POD [41]. The preoperative usage of medications such as propranolol, scopolamine, and benzodiazepines was a strong predictor of POD. The study noted that these patients were at a 12-fold higher risk of developing POD than those who were not exposed to psychoactive medications or polypharmacy preoperatively.

Screening tools that detect preoperative cognitive impairment have also been used to predict POD. A review article analyzed available cognitive screening tools for identifying preoperative mild cognitive impairment. It found that the Montreal Cognitive Assessment (MoCA) had the highest sensitivity and specificity to detect mild cognitive impairment (81–93% and 74–89%, respectively) [42]. It should be noted that if cognitive impairment is detected preoperatively, patients should be referred to a geriatrician or neurologist for further comprehensive neuropsychologic evaluation if feasible prior to surgery [43]. These evaluations may lead to primary or secondary diagnoses of mild cognitive impairment such as Alzheimer’s or Parkinson’s Disease, the latter of which would benefit from dopamine agonists and cholinesterase inhibitors [44].

### 4.2. Intraoperative Interventions

Several intraoperative strategies may potentially further reduce the likelihood of cognitive impairment after cardiac surgery. Not surprisingly, the duration of a patient on CPB plays a role in the development of POD. In a 2022 dual center study, there was a correlation between longer duration of CPB with an increased risk of developing delirium. Of the patients who developed delirium, the median CPB time was 18 min longer when compared to the patients which did not. CPB time is often linked to the type of procedure, where combined coronary artery bypass grafting (CABG) and valve surgery demand longer durations. Per this study, patients undergoing combined surgery more frequently developed post-operative delirium compared to CABG or valve alone [45].

Several physiologic changes occur during cardiac surgery, each of which may contribute to the development of delirium. Patients can experience rapid changes in temperature, intravascular pH and arterial pressure. These changes affect a patient’s cerebral perfusion, oxygen extraction, and oxygen consumption. Because of this, brain tissue may experience hypoxia and/or ischemia [46]. Thus, utilizing cerebral oximetry may help reduce the incidence of delirium by measuring cerebral oxygenation. A 2017 randomized control trial measured pre-operative and intraoperative cerebral oxygenation saturation via cerebral oximetry during cardiac surgery. The study noted a four-fold increase in POD in patients that experienced cerebral oxygenation less than 50% of baseline intraoperatively [47].

One would expect that an appropriate response to cerebral hypoxia is to increase the oxygen concentration provided to the patient. However, Lopez et al. noted POD with hyperoxic reperfusion following hypoxia [46]. In this cohort study, 51% of patients undergoing cardiac surgery developed cerebral ischemia (defined as cerebral oxygenation less than 80% from baseline for over 5 min). A total of 94 of these patients experienced hyperoxia (cerebral oxygenation above baseline) after interventions were made to correct the ischemia. Despite the correction of cerebral ischemia, 40% of the patients experiencing hyperoxia developed POD. POD was also associated with hyper-oxygenation independent of cerebral ischemia. The suggested mechanisms for such findings include oxidative injury and/or hyperoxic vasoconstriction. Irrespective of the cause, these findings challenge evidence supporting perioperative hyper-oxygenation to reduce the risk of POD [46].

Another important intraoperative factor affecting the incidence of POD is the depth of anesthetic used. A general anesthetic is often required during cardiac surgery. However, POD is a known complication of general anesthesia, even in non-cardiac surgeries. The development of the Bispectral Index monitor (BIS) has allowed for a more tailored anesthetic. The BIS monitor measures EEG information and interprets it as a number representing anesthetic depth [48]. Santarpino et al. incorporated the use of a BIS monitor during aortic surgeries to find associations between intraoperative EEG changes and POD. Measurements taken after anesthetic induction were used as baseline values. Their data showed increased POD risk with a BIS decrease of 25–30% from baseline [48]. A direct cause cannot be made, but the use of BIS has value when attempting to minimize risk of POD.

While the BIS monitor is designed to measure the depth of anesthesia provided, it has been used to indirectly monitor cerebral blood flow as well. Liu et al. designed an observational study to investigate whether cerebral perfusion contributes to low BIS values [49]. The researchers demonstrated an association between a low BIS value and incidence of POD. Moreover, a decrease in cerebral perfusion would result in an increase in slow wave EEG. If the reduction in perfusion continued, the EEG would go silent [49]. While this study did not aim to argue the importance of anesthetic depth monitor, it did highlight the BIS as a monitor which may be able to identify other factors affecting the risk of POD.

Several pharmacological strategies have been trialed to prevent and treat delirium, with various degrees of success [20]. Many therapies are aimed at improving circadian rhythm and avoiding sleep disturbances in the intensive care unit (ICU) setting. Dexmedetomidine is one such agent that has become an intriguing option in addressing POD. When compared to propofol for sedation, dexmedetomidine has been shown to have a reduced incidence of POD in the ICU setting. This is believed to be due to its reduction in opioid requirements, improved quality of sleep, and reduction in inflammatory response [50]. There was also the added benefit of reducing the extubation time, as dexmedetomidine did not cause significant respiratory depression [50]. However, the effectiveness of dexmedetomidine as a preventive intervention remains inconclusive. A 2017 double-blind, randomized controlled trial assessed the incidence of delirium after elective cardiac surgery when dexmedetomidine was infused intraoperatively as compared to normal saline infusion. Of the 285 patients in the study, only 5% developed delirium [51]. This was not significantly different from the 7.7% that developed delirium with a normal saline infusion [51].

Ketamine, a dissociative anesthetic, is often associated with emergence delirium [52]. Interestingly, a meta-analysis on commonly used medications during cardiac found ketamine may have protective properties against POD. Patients receiving 0.5 mg/kg of intravenous ketamine on anesthetic induction showed an approximately thirteen times reduced incidence of POD compared to placebo [52]. The most obvious possibility being its ability to reduce opioid requirements both intra- and post-operatively. However, an alternate hypothesis is ketamine serving as a neuroprotective agent. Patients receiving ketamine have lower C-reactive protein levels when compared to placebo. Thus, ketamine’s association with the reduction in POD may be due to its anti-inflammatory property regarding brain mechanics.

### 4.3. Postoperative Interventions

Delirium in the postoperative setting of cardiac patients poses a significant issue in terms of recovery. Due to its multifaceted and complex etiology, it continues to be underdiagnosed and mismanaged. The aging population, and thus the consequential aging nature of the cardiac patient, has increased the prevalence of this issue in the cardiac surgical population [4]. Risk factors for postoperative delirium have been identified as postoperative stroke or transient ischemic attack, prolonged mechanical ventilation for a duration of longer than 24 h, patients older than 65, surgical interventions of CABG or valve surgery, postoperative blood product transfusion, postoperative renal insufficiency, or prehospital benzodiazepine use [53].

For lessened or diminished effects of these risk factors, postoperative interventions have been suggested for the reduction or prevention of delirium. Nonpharmacologic interventions such as reorientation, sleep protocols, the promotion of early mobility, avoidance of restraints, adequate nutrition and hydration, adequate oxygenation, appropriate medication usage, appropriate pain management, and visual and hearing aids have all been shown to decrease the incidence of postoperative delirium or mitigate the effects [53,54,55,56,57]. In one single-blind, randomized controlled trial on 114 patients who underwent CABG, patients were randomized to routine care plus eye masks and earplugs during nighttime as compared to patients who received only routine postoperative care. The severity of POD was significantly reduced in the experimental group, while having positive effects on overall sleep quality [58].

Evidence for the use of antipsychotic pharmacologic intervention to lessen postoperative delirium in patients is limited. The usage of risperidone, olanzapine, and haloperidol are all accepted for the treatment of delirium [52]. Risperidone has been tested in the setting of postoperative recovery after cardiac surgery, and was found to significantly reduce the incidence of postoperative delirium [59].

Melatonin and ramelteon have also been used for reducing post-operative delirium. Melatonin, an endogenous hormone originating from the pineal gland, not only plays a pivotal role in regulating the circadian sleep–wake cycle, but also acts as a free radical scavenger, exerting anti-inflammatory effects [60,61]. Ramelteon is a melatonin agonist. Evidence of its efficacy is conflicting, however. A meta-analysis of randomized controlled trials on the use of melatonin and ramelteon, as compared to placebo, on delirium in the ICU found no significant reduction in delirium among patients [60]. Concerning patients undergoing cardiac surgery, a meta-analysis examining the influence of melatonin on POD indicated that perioperative melatonin administration significantly mitigates POD in this patient population. However, this conclusion was based on a low level of evidence from the included studies [61].

As mentioned, systemic inflammatory response from surgery and CPB contributes to POD. Minocycline is a tetracycline derivative that can act as an anti-inflammatory agent and has been investigated as to whether it can decrease the risk of developing POD. In rat models undergoing CPB, the postoperative administration of minocycline has been demonstrated to decrease the activation of the inflammatory microglia cells while increasing cell neurogenesis, thus attenuating postoperative cognitive deficits [62]. However, in a randomized control trial, 100 patients undergoing total knee arthroplasty under general anesthesia were randomized to receiving minocycline one day preoperatively and for seven days post-operatively [63]. The researchers found no difference between the types of prevention of postoperative delirium. Further research would need to be conducted in human trials to assess the impact of perioperative administration of minocycline on the prevention of POD.

## 5. Enhanced Recovery after Surgery

Enhanced Recovery After Surgery (ERAS) bundles emerged in the late 1990s as a multimodal perioperative care pathway designed to achieve optimized recovery following surgical procedures by preserving preoperative physiological functionality and reduction in the natural stress response after surgical intervention [64]. ERAS protocols provide a standardized yet individualized methodology of compromising patient care modalities together with evidence-based practices at their root. These bundles are dynamic as evidence-based medicine progresses; however, their main postoperative components consist of targeted early chest drain management, early extubation, early ambulation, goal-directed therapy, multimodal analgesia, and thromboprophylaxis (Enhanced Recovery After Surgery Cardiac Society, 2023). The ERAS Cardiac Society outlines within their recommendations that delirium screening is an important component of postoperative recovery. Reports currently suggest that up to 20% of cardiac surgery patients experience delirium during their hospital course [65,66]. Their recommendations are based on varying levels of evidence-based practices and the current available literature, which can be found on the ERAS Cardiac society’s website with varying levels of recommendation and evidence.

## 6. Conclusions

POD in cardiac surgical patients can lead to poor functional outcomes for patients. It increases the hospital length of stay and costs, and is associated with longer-term cognitive dysfunction. There are many modifiable and non-modifiable risk factors that predispose patients to POD, including dementia, depression, hypertension, diabetes, history of stroke, or transient ischemic attack (TIA). Systemic inflammation from surgery and CBP may also significantly impact neurocognitive outcomes. Interventions to mitigate the development of POD lie within addressing patient-specific modifiable risk factors, targeting neuroinflammatory pathways, enhancing intraoperative monitoring and management, and optimizing the postoperative care of these patients.

## Data Availability

No new data were created or analyzed in this study. Data sharing is not applicable to this article.

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
