# Peer review of "Delirium after Cardiac Surgery—A Narrative Review"

_brainsci, 2023, doi:10.3390/brainsci13121682_

Round 1

Reviewer 1 Report

Comments and Suggestions for Authors

I suggest to improve the context of the sentence "...substantial reduction in the incidence of POD remains high among cardiac surgical patients.", in the abstract.

Introduction, page 2

...Obtaining an appropriate history and physical (examination) can...

Searching Strategy

...case reports, systematic reviews...

Management of Postoperative Delirium...

...For example, blood pressure management during CPB is typically kept at a minimum of 50 mmHg to maintain...

What blood pressure? Mean arterial pressure?

Author Response

Please see the attached word document. 

Reviewer 2 Report

Comments and Suggestions for Authors

I read with great interest the review by Mattimore et. al on delirium after cardiac surgery. The manuscript is interesting and sound. However, there are some issues that need to be addressed:

- Why did you include articles from 1987 on? What is the meaning of this particular year used as a cut-off? Please explain.

- In the search strategy, authors should specify which outcomes they were focused on, and whether they looked for some particular intervention or comparison, in order to clarify the screening strategy. Please also specify how the screening was conducted.

- In part 2, authors should briefly discuss anticoagulation for extracorporeal circulation and report that new anticoagulation strategies have been tested in order to reduce intraoperative and postoperative complications (doi: 10.1016/j.thromres.2022.02.007 - doi: 10.1186/s13019-023-02359-2 - doi: 10.1053/j.jvca.2011.09.002), although more research is warranted. Please discuss and add these 3 references.

- When talking about preoperative interventions, authors should also discuss the use of melatonin (doi: 10.5005/jp-journals-10071-24571) and ramelteon (doi: 10.3390/jcm12020435) as pharmacological strategies as well as the possible use of non-pharmacological strategies (doi: 10.1016/j.aucc.2023.08.003). Please discuss and add these 3 references.

Author Response

Please see the attached word document

Reviewer 3 Report

Comments and Suggestions for Authors

Review: Delirium after cardiac surgery:

1. The introduction is too detailed and should start with the main topic of the paper, which is POD. The CPB procedure is explained to extensively as the focus of the paper is not CPB, but a review of POD risk and management factors.

2. To my knowledge POD appears days after surgery and could last for a long time, but is unlikely to appear weeks after surgery. For this statement the authors should cite an appropriate paper.

3. The authors focus very much on the explanation of CPB. As this paper is meant to be a review it should not focus on the details of cardio-surgical procedure or other details that are not necessary to deepen the knowledge about POD in cardiac patients.

4. There is some redundant and unnecessary information in section 2 about CBP.

5. Abbreviations should be explained when mentioned for the first time.

6. There are some sentences that are quite difficult to understand and sentences where some information or words are missing, for example:

·        Despite our greater understanding of these causative factors, substantial reduction in the incidence of POD remains high among cardiac surgical patients.

·        Obtaining an appropriate history and physical can prospectively identify patients at highest risk,…..

·        Intraoperatively, certain factors have been proposed to increase or predict the risk of post-operative neurocognitive disorders. These include CPB time, surgical complexity, and use of a cerebral oximetry or bispectral index (BIS).

·        One could think, that using BIS is a risk factor for POD. Maybe it would be better to split this sentence.

·        Release of interleukin 6 (IL-6) and C-reactive protein are involved in the inflammatory cascade and lead to increased permeability of the blood-brain-barrier can lead to cerebral edema and increased activity of neuro-immune cells such as microglial cells.

·        Pulsatile blood flow is controversial to whether any significant benefit exists.

·        The molecular mechanisms ischemia-reperfusion injury has been well studied in hopes that targeting involved pathways with new therapies may reduce cerebral injury.

·        Postoperatively, non-pharmacological interventions such as reorientation, good sleep hygiene, early mobility, visual and hearing optimization, adequate nutrition, and cognitive exercise have resulted in a dose-dependent response in the reduction rates of postoperative delirium.

·        Reperfusion of these cells exposed to ischemia may hasten cell death by introducing oxygen-derived free radicals, disrupting the blood brain barrier and increasing inflammatory mediators such as cytokines, TNF-alpha, IL-1 beta, IL-6 from microglial activation.

·        This incidence of this has been reported to be as high as >50%........

·        A recent metanalysis with a pooled >60,000 patients who underwent CABG noted modifiable risks factors have been associated with POD.

·        Multimodal magnetic resonance imaging following CPB has suggests potential brain infarction in up to 45% of patients following cardiac surgical interventions.

·        Liu et al, created an observational study to investigate whether a low BIS value accounts only for the depth of anesthesia or if cerebral perfusion is a factor.

·        While this study saw the association between a low BIS and incidence of POD, this association continued when end-tidal anesthetic concentration remained the same across patients.

·        Several physiologic changes occur during cardiac surgery which play independent risk factors for the development of delirium.

·        This suggests a significant intraoperative change from baseline carries a better predictive value than a relative change.

·        Wide-spread inflammation from surgery, CBP may significantly impact neurocognitive outcomes.

·        Several pharmacological strategies have been trialed to prevent and treat delirium with wavering success [38]. Many often involved improving circadian rhythm and avoiding sleep disturbances in the ICU setting. Therefore, dexmedetomidine has become an intriguing option in addressing POD.

·        A 2017 randomized double-blind controlled trial assessed the incidence of delirium after elective cardiac when dexmedetomidine was infused intraoperatively.

·        Fortunately, the study did not find any increase in incidence which favors the possibility for larger studies.

7. The section about minocycline should be put in the postoperative chapter, as it is a postoperative measure.

These sentences seem to be grammatically incorrect:

Postoperative administration of minocycline has been demonstrated in rat models undergoing CPB to decrease activation of the inflammatory microglia cells, increases cell neurogenesis, and attenuates cognitive deficits.

Further research would need to be conducted in human trials to assess the impact of pre- or postoperative administration of minocycline on POD.

8. There are contradicting statements like:

Few interventions have proven to be reliable tools for screening, preventing or treating delirium.

And one page later the authors state:

Nonpharmacologic interventions such as reorientation, sleep protocols, promotion of early mobility, avoidance of restraints, adequate nutrition and hydration, adequate oxygenation, appropriate medication usage, appropriate pain management, and visual and hearing aids all have shown to decrease incidence of postoperative delirium or mitigate the effects

9. There are quite a lot of redundant parts. Even though some aspects may be seen in more than one context it would be better not to repeat unnecessary details.

Comments on the Quality of English Language

Every sentence should contain one or two matching pieces of information. If you put too many different aspects in one sentence it could get very confusing.

Author Response

Please see attached word document

Round 2

Reviewer 3 Report

Comments and Suggestions for Authors

There are still some issues that should be addressed in the review:

1. The structure of the review should follow the PRISMA checklist for Reviews as stated in

the guidelines for authors.

2. There are still some abbreviations, that have not been explained (ICU, TIA).

3. There is a spelling error on page 3 first paragraph.

4. It seems that there is a verb missing in this sentence:

Emboli may be in the form of atherosclerotic calcium deposits dis-lodged from aortic arch cross clamping, or gaseous microemboli resulting from entrapped air in the CPB circuit (8, 9).

In regard to patients undergoing cardiac surgery, a meta-analysis investigating the impact of melatonin on POD suggested perioperative use of melatonin significantly POD in patients undergoing cardiac surgery.

5. In general, as this is a review, the authors should cite the relevant paper for their statements, for example in these sections:

No exact pathophysiological change responsible for POD is known but there are multifaceted etiologic factors contributing to its occurrence. In cardiac sur-gery, factors such as systemic inflammation, hypothermia, non-pulsatile flow from CPB and ischemic-reperfusion injury may all contribute to POD.

Brain injury is a major factor of postoperative morbidity after CPB and is as-sociated with prolonged hospitalization, excessive operative mortality, increased hospital costs, and decreased quality of life.

Melatonin is a naturally occurring hormone produced from the pineal gland and is not helps to regulate the circadian sleep-wake cycle but is a free radial scavenger and exerts anti-inflammatory effects. (there is also a spelling mistake in this sentence.)

Comments on the Quality of English Language

The English is good and quite easy to read.
